# Exploring a Novel Role of Glycerol Kinase 1 in Prostate Cancer PC-3 Cells

**DOI:** 10.3390/biom14080997

**Published:** 2024-08-13

**Authors:** Bobae Park, Sang-Hun Kim, Sun-Nyoung Yu, Kwang-Youn Kim, Hoyeon Jeon, Soon-Cheol Ahn

**Affiliations:** 1Department of Molecular Medicine, University of Texas Health San Antonio, San Antonio, TX 78229, USA; parkb@uthscsa.edu; 2Department of Microbiology & Immunology, Pusan National University School of Medicine, Yangsan 50612, Republic of Korea; gurisn@naver.com (S.-N.Y.); hoyhoy1994@hanmail.net (H.J.); 3Section of Pulmonary, Critical Care and Sleep Medicine, Department of Internal Medicine, Yale University School of Medicine, New Haven, CT 06520, USA; sang-hun.kim@yale.edu; 4Korean Medicine (KM)-Application Center, Korea Institute of Oriental Medicine (KIOM), Daegu 41062, Republic of Korea; lokyve@kiom.re.kr

**Keywords:** apoptosis, glycerol kinase, PC-3 cells, tumor suppressor, exogenous expression

## Abstract

Clinically, prostate cancer is infamous for its histological and molecular heterogeneity, which causes great challenges to pinpoint therapy and pharmaceutical development. To overcome these difficulties, researchers are focusing on modulating tumor microenvironment and immune responses in addition to genetic alteration and epigenetic regulation. Here, we aimed to identify potential biomarkers or modulators of prostate cancer by investigating genes specifically altered in prostate cancer cells treated with established anti-cancer agents. Glycerol kinase 1 (GK1) is phosphotransferase encoded on the X chromosome, is associated with the synthesis of triglycerides and glycerophospholipids, and has been mainly studied for X-linked metabolic disorder GK deficiency (GKD). Interestingly, our DNA microarray analysis showed that several anti-cancer agents highly induced the expression of GK1, especially GK1a and GK1b isoforms, in human prostate cancer PC-3 cells. To elucidate the relationship between GK1 and cancer cell death, a human GK1b-specific expression vector was constructed and transfected into the PC-3 cells. Surprisingly, GK1b overexpression dramatically reduced cell viability and significantly accelerated apoptotic cell death. These findings suggest that GK1b may serve as a promising modulator and biomarker of cell death in prostate cancer, offering potential avenues for therapeutic intervention.

## 1. Introduction

Prostate cancer is the most complicated and frequently diagnosed disease among men. In 2023, its incidence and mortality rates held the top positions in the United States, with 29% (288,300 estimated new cases) and 11% (34,700 estimated deaths), respectively [1,2]. Generally, the early detection of prostate cancer increases the likelihood of effective treatment and decreases the recurrence rate, resulting in a 5-year relative survival rate of 97% [1]. Nonetheless, even with positive outcomes from early intervention, 20 to 50% of men experience recurrence within a decade of initial definitive treatment [3,4,5]. The biggest challenge in managing prostate cancer is its heterogeneity, which makes accurate diagnosis and treatment problematic. This heterogeneity is influenced by genetic and non-genetic factors [6,7], including genetic alterations, epigenetic modifications, and a tumor microenvironment that promotes unrestricted cell proliferation and invasiveness [7,8]. Genetic alterations that partially change the DNA sequence involve mutation, genomic instability, loss of heterozygosity (LOH), and gene copy number variation (CNV), whereas epigenetic modifications occur through histone modifications, DNA methylation, and loss of imprinting (LOI) without altering the DNA sequence [8,9].

Glycerol kinase (GK) is well known as a mediator of carbohydrate and lipid metabolism. The GK family has been identified into four distinct subtypes: *GK1*, *GK2*, *GK3P*, and *GK5*. *GK1* is further subdivided into four isotypes: isoforms a, b, c, and d. Surprisingly, GK subtypes are mapped on different human chromosomes: *GK1* on X chromosome, *GK2* and *GK3P* on chromosome 4, and *GK5* on chromosome 3 (Table 1). Basically, GK activity in white adipocytes (WATs) is negligible, which explains why WATs cannot metabolize the glycerol obtained during the degradation of triacylglycerol. This glycerol is transferred to the liver through blood and used as a source for glycolysis or gluconeogenesis in sequential enzymatic reactions. The most studied clinical area on GK is the GK deficiency (GKD), which is also known as hyperglycerolemia. GKD is an X-linked (loss of the Xp21 region) glycerol metabolic disease biochemically characterized by the accumulation of glycerol in blood and urine [2,10]. Rahib et al. have reported that GKD alters expressions of genes involved in lipid metabolism, carbohydrate metabolism, and insulin signaling [11]. This suggests that GK not only regulates carbohydrate and lipid metabolism but also insulin resistance. Meanwhile, unlike previous reports, Sriram et al. have discovered that GK-overexpressing rat hepatoma consumes carbon sources at higher rates [12]. In addition, Lounis et al. have reported that prostate tumors express elevated levels of glycerol-3-phosphate (Gro3P) phosphatase (G3PP), suggesting that G3PP is an important biomarker for prostate cancer [13]. They speculated that the high expression of G3PP in cancer cells may have contributed to survival in a toxic environment induced by nutrient excess. G3PP hydrolyzes Gro3P to glycerol, which reacts in the opposite way to GK. In other words, GK overexpression could potentially lead to excessive carbon consumption and glucolipotoxicity in prostate cancer, disrupting intracellular energy balance and eventually triggering apoptosis.

Interestingly, our DNA microarray analysis confirmed a significant increase in the gene expression of GK in prostate cancer PC-3 cells treated with several anti-cancer agents, suggesting the involvement of the GK gene in cancer cell death.

This study proposes that GK may have a novel function in prostate cancer, and it investigates the impacts of exogenously transfected human GK on prostate cancer cells. Prior to investigating the novel role of GK in prostate cancer, we constructed a recombinant human GK plasmid to be expressed in mammalian cells. Furthermore, RNA and protein production assays, GK activity analysis, subcellular fractionation, cell viability assays, and apoptosis analysis were executed to determine whether GK has anti-tumor activity in prostate cancer cells.

## 2. Materials and Methods

### 2.1. DNA Microarray Analysis

A Whole Human Genome Microarray (4 × 44 K) kit (Agilent, Santa Clara, CA, USA) was used to analyze gene expression alterations after human prostate cancer PC-3 cells were treated with several anti-cancer agents, such as compound K, platycodin D, lasalocid, deoxypodophyllotoxin, piplartine, salinomycin, and pipernonaline. Agilent 2100 Bioanalyzer was used rather than conventional gel electrophoresis to evaluate the quality of isolated RNAs. The experimental sequence involved (i) sample preparation (labeling), (ii) hybridization, and (iii) microarray wash using the Low RNA Input Linear Amplification kit PLUS, the Gene Expression Hybridization Kit and the Gene Expression Wash Buffer Kit, respectively (all purchased from Agilent). Cy3-labled and Cy5-labled target complementary RNAs (cRNAs) (750 ng of each) were combined and amplified. Then, cRNA was fragmented by treating with a 10× Gene Expression Blocking Agent (Agilent) and 25× Fragmentation Buffer (Agilent) for 30 min at 60 °C, immediately cooled on ice, and the reaction was stopped by adding 2× Hybridization Buffer. Samples were loaded directly into an assembled chamber in an Agilent hybridization oven for 17 h at 65 °C. The microarray chamber was washed according to the manufacturer’s instructions. Microarray analysis was conducted using GenomicTree (Daejeon, Republic of Korea). And raw data were normalized and clustered by gene fold-changes using GebeSpringGX 7.3.1 (Agilent). Mean normalized ratios were calculated by dividing the mean intensities of normalized signal channels by the mean intensities normalized control channels. More than 2–fold changes were regarded as meaningful genes. Functional annotations of genes were analyzed using the DAVID bioinformatics resources 6.8 web accessible program “https://david.ncifcrf.gov (accessed on 13 June 2024)”, which is a database for annotation, visualization, and integrated discovery.

### 2.2. Construction of Human Glycerol Kinase 1 Isoform b (GK1b) Expression Vector

A schematic diagram of the construction of the GK1b expression vector is shown in Figure 1A. Briefly, full-length human GK1 mRNA containing an extra insertion of a HindIII restriction enzyme site was synthesized and amplified from the genomic DNA of human prostate cancer PC-3 cells using a 5× Reverse Transcription Premix (ELPIS, Daejeon, Republic of Korea) and Taq polymerase (Takara, Shiga, Japan) kit. The following primer sequences were used: full-length human GK1-HindIII forward *5*′-GAAGCTTATGGCAGCCTCAAAGAAG-*3*′ and reverse *5*′-GAAGCTTTTATGGAATACCACTTTCTG-*3*′. Amplified GK1-*5*′- and *3*′-HindIII overhang fragments were ligated into a pGEM^®^-T Easy vector (Promega, Madison, WI, USA). To establish a mammalian expression vector containing exogenous human GK1 genes, sub-cloning was executed with a pcDNA3.1/Hygro(+) vector (Addgene, the nonprofit plasmid repository, Watertown, MA, USA). Recombinant pGEM^®^-T Easy/GK1 (donor plasmid) and pcDNA3.1/Hygro(+) (recipient plasmid) were digested with HindIII enzymes (Enzynomics, Daejeon, Republic of Korea), ligated using the T4 ligase system (Enzynomics), and transformed into DH5α competent cells. The nucleotide sequence of the recombinant plasmid DNA was confirmed by a DNA sequencing analyzer (Cosmo Genetech, Daejeon, Republic of Korea) and identified as GK1b (Figure 1B). A plasmid midi kit (QIAGEN, Hilden, Germany) and EndoZero II Spin-Column (Zymo Research, Irvine, CA, USA), which reduces endotoxin levels for plasmid DNA from ≤1 EU/μg to ≤0.025 EU/μg, were used to obtain large quantities of recombinant plasmids. For further experiments, the empty vector (pcDNA3.1/Hygro(+)) was named ‘Mock’ and the recombinant GK1b vector (pcDNA3.1/Hygro(+)/GK1b) was named ‘GK1b’.

### 2.3. Cell Lines, Cell Culture, and Transfection of Human GK1b Expression Vector

Human prostate cancer cell lines PC-3, DU-145, and LNCaP, human breast cancer cell lines MCF7 and MDA-MB-231, human glioblastoma cell line U251MG, murine colorectal cancer cell line CT26, human colorectal cancer cell line HCT116, and hepatocellular carcinoma cell line HepG2 were acquired from the American Type Culture Collection (ATCC). LNCaP, MDA-MB-231, CT26, and HCT116 cells were grown in Roswell Park Memorial Institute Medium (RPMI) 1640 medium (Sigma-Aldrich, St. Louis, MO, USA) supplemented with 10% fetal bovine serum (FBS) and 1% penicillin/streptomycin solution at 37 °C and 5% CO_2_. Another cell line was cultured in Dulbecco’s Modified Eagle Medium (DMEM) (Sigma-Aldrich) supplemented with 10% fetal bovine serum (FBS) and 1% penicillin/streptomycin solution at 37 °C and 5% CO_2_.

PC-3 cells were transfected using OmicsFect transfection reagent (OmicBio, New Taipei City, Taiwan), according to the manufacturer’s instructions. Briefly, cells were seeded onto a 6-well culture plate and incubated for 24 h. An OmicsFect transfection reagent in a serum-free medium (mixture) was incubated at room temperature for 5 min. Subsequently, DNA was directly added to the mixture (DNA-OmicsFect complex) and incubated at room temperature for 30 min. The culture media was then replaced with fresh DMEM containing 10% FBS without antibiotics, and the DNA-OmicsFext complexes were carefully added dropwise to cells and incubated with gentle shaking at 37 °C for 48 h.

### 2.4. RNA Isolation and RT-qPCR Analysis

The total RNAs were extracted from PC-3 cells using a RiboEx^TM^ and Hybrid-R^TM^ kit (GeneAll, Seoul, Republic of Korea) according to the manufacturer’s instructions. Complementary DNAs (cDNAs) was synthesized from 1 μg of total RNAs by incubating for 60 min at 37 °C with 5× Reverse Transcription Premix (ELPIS), then increasing the temperature to 94 °C and incubating for 5 min to stop the reaction. Quantitative PCR (qPCR) was conducted using a Real-Time qPCR 2× Master Mix (ELPIS) and an ABI QuantStudio3 machine (Applied Biosystems, Waltham, MA, USA). The following cycling conditions were performed: 94 °C for 3 min, followed by 40 amplification cycles of 94 °C for 20 s, 58 °C for 20 s, and 72 °C for 20 s. The DNA melting curve analysis was examined by increasing the temperature from 60 to 95 °C at 0.5 °C/s. The comparative CT method (calculated by 2^−ΔΔCt^) was used to analyze the relative expression of human GK1 isoform mRNAs. The primers used were GK1 isoform a (GK1a) (accession number NM_203391.4) forward *5*′-TGCAAGTAGGACTATGCTTT-*3*′ and reverse *5*′-CAGCTTTCACGCTATGAGA-*3*′; human GK1 isoform b (GK1b) (accession number NM_000167.6) forward *5*′-TATGGCCTAATGAAAGCTGG-*3*′ and reverse *5*′-AATCTGGAAGCACATTTGTC-*3*′.

### 2.5. Western Blotting

Proteins were extracted from cells with a PRO-PREP^TM^ Protein Extraction Solution (iNtRON Biotechnology, Seongnam, Republic of Korea), containing phosphatase inhibitor cocktail 3 (Sigma-Aldrich) and phenylmethylsulfonyl fluoride (PMSF) (Sigma-Aldrich) [16]. Proteins were quantified using a BCA Assay Kit (iNtRON Biotechnology) according to the manufacturer’s instructions. The primary antibodies used were glycerol kinase (Abcam, Cambridge, UK; Cat#ab126599; dilution 1:2000), PARP (Cell signaling technology (CST), Danvers, MA, USA; Cat#9542; dilution 1:2000), Caspase-3 (CST; Cat#9662; dilution 1:1000), Bax (Santa Cruz Biotechnology (SCBT), Dallas, TX, USA; Cat# sc-7480; dilution 1:2000), BCL-2 (CST; Cat#2872; dilution 1:1000), and β-Actin (SCBT; Cat#sc-47778; dilution 1:4000). The secondary antibodies used were horseradish peroxidase (HRP)-conjugated goat-anti-mouse IgG (Enzo Life Science (Enzo), Farmingdale, NY, USA; Cat#ADI-SAB-100-J; dilution 1:4000) and goat-anti-rabbit IgG secondary antibodies (Enzo; Cat#ADI-SAB-300-J; dilution 1:4000). WestGlow^TM^ FEMTO ECL solution (BIOMAX, Seoul, Republic of Korea; Cat# BWF0100) was used to detect picogram amounts of protein. Western blot signals were obtained using a Bio Molecular Imager System (IQ800, Amersham, Amersham, UK) and quantified using ImageJ 1.54 “https://imagej.net/ij/ (accessed on 13 June 2024)”. Original figures can be found in Appendix A.

### 2.6. Measurement of GK Activity

To examine universal GK enzyme activity, PC-3 cells were transfected with either Mock or recombinant human GK1b plasmid, incubated for 48 h at 37 °C, and gently washed with twice with ice-cold 1× phosphate-buffered saline (PBS). Cells were harvested by scraping and centrifuged at 3000 rpm for 4 min at 4 °C. GK activities were determined using a Glycerol-3-Phosphate (Gro3P) Colorimetric Assay Kit (BioVision, Milpitas, CA, USA). Briefly, cell pellets were suspended in Gro3P Assay buffer and immediately homogenized. Lysates were collected by centrifugation at 13,000 rpm for 10 min at 4 °C. Protein concentrations in lysates were determined using the BCA method. Lysates were then gently mixed with a Gro3P Probe and Gro3P Enzyme Mix and incubated for 1 h at 37 °C in the dark. Colorimetric analysis was conducted using an enzyme-linked immunosorbent assay (ELISA) reader (Molecular Devices, San Jose, CA, USA) at 450 nm.

### 2.7. Preparation of Subcellular Fractionation

This procedure was conducted as described by Baghirova et al. [17] with modification. Briefly, PC-3 cells were seeded at a confluence of 80–90% into 100 mm cell culture dishes and transfected with either Mock or recombinant human GK1b plasmid for 48 h. After that, the medium was removed, and cells were washed with 1× PBS and harvested by centrifugation at 500× *g* for 10 min at 4 °C. Cell pellets were lysed with ice-cold lysis buffer A [150 mM NaCl, 50 mM HEPES (pH 7.4), 25 μg/mL digitonin and 1M hexylene glycol] containing protease inhibitor and 1% phosphatase inhibitor cocktail. The pellets obtained by centrifugation were resuspended by vortexing and incubated on an end-to-end rotator for 10 min at 4 °C; then, they were centrifuged at 2000× *g* for 10 min at 4 °C. At this point, the supernatant fractions contained cytosolic proteins. Pellets were mixed with ice-cold lysis buffer B [150 mM NaCl, 50 mM HEPES (pH 7.4), 1% (*v*/*v*) Igepal and 1 M hexylene glycol] containing a protease inhibitor and 1% phosphatase inhibitor cocktail. Pellets obtained by centrifugation were resuspended by vortexing and incubated on ice for 30 min; then, they were centrifuged at 7000× *g* for 10 min at 4 °C. These supernatant fractions included membrane-bound organelles, such as mitochondria, endoplasmic reticulum and Golgi apparatus. To obtain nucleus fractions, the remaining pellets were suspended with ice-cold lysis buffer C [150 mM NaCl, 50 mM HEPES (pH 7.4), 0.5% (*w*/*v*) sodium deoxycholate, 0.1% (*w*/*v*) sodium dodecyl sulfate, and 1M hexylene glycol] containing a protease inhibitor, 1% phosphatase inhibitor cocktail and benzonase and incubated on an end-to-end rotator for 30 min at 4 °C. The suspensions were then centrifuged at 7800× *g* for 10 min at 4 °C. These supernatant fractions included nuclear proteins.

### 2.8. Cell Viability Assay and Annexin V/Propidium Iodide (PI) Analysis

Cell viability was estimated by using the MTT [3-(4,5-Dimethylthiazol-2-yl)-2,5-Diphenyltetrazolium Bromide] assay. Cells were transfected with either Mock or recombinant human GK1b plasmid and incubated for 72 h at 37 °C. An equal volume of 1 mg/mL MTT solution was then added to the medium and incubated for 3 h at 37 °C. The insoluble formazan crystals were dissolved in dimethyl sulfoxide (DMSO, Junsei, Tokyo, Japan), and colorimetric analysis was performed using an ELISA reader at 570 nm. To determine whether exogenous human GK1 influences cell death including apoptosis, PC-3 cells were transfected, as described above, and incubated for 48 h at 37 °C. A FITC Annexin-V apoptosis Detection kit (BD Bioscience, Franklin Lakes, NJ, USA) was used to quantify apoptotic cell death. Briefly, post-transfection, cells were harvested by trypsinization and washed with 1× PBS. Collected cells were then mixed with 1× binding buffer and stained with an FITC annexin-V and PI at room temperature for 15 min in the dark. Fluorescence intensities were measured using a FACS CANTO II (BD Bioscience) and quantified using FlowJo 10.10 (FLOWJO, Ashland, OR, USA).

### 2.9. Statistics

All experiments were repeated at least three times and produced consistent results. Unless otherwise stated, data are expressed as means ± standard error of the means (SEMs). One-way analysis of variance (ANOVA) was used to compare experimental groups to control values, and multiple groups were compared using Tukey–Kramer multiple comparison tests. Statistical significance was accepted for *p* values * < 0.05, ** < 0.01, or *** < 0.001, as indicated.

## 3. Results

### 3.1. Exploring Novel Biomarkers by DNA Microarray Analysis

To investigate novel biomarkers, DNA microarray analysis was conducted on human prostate cancer PC-3 cells treated with compounds derived from natural substances, including compound K, platycodin D, lasalocid, deoxypodophyllotoxin (DPT), piplartine, salinomycin, and pipernonaline. Previous studies have shown that these compounds have powerful anti-cancer properties [18,19,20,21,22,23,24]. We found that the expressions of many genes were changed by these agents compared to the untreated group. Human glycerol kinase (GK)1 genes, especially GK1 isoform a (GK1a) and GK1 isoform b (GK1b), were upregulated by >2-fold (Table 2). To verify DNA microarray results, PC-3 cells were treated with three anti-cancer agents randomly selected, such as piplartine, deoxypodophyllotoxin (DPT), salinomycin, used in the microarray and one additional agent, compound PN [25]. The expression levels of the GK1 mRNAs, particularly its GK1a and GK1b isoforms, were significantly and dose-dependently increased after treatment with four anti-cancer agents (Figure 2A). These results were also consistent with the effect on protein levels (Figure 2B).

Next, we sought to determine whether GK1 is widely expressed in various cancer cell types. We found that endogenous GK1 was markedly expressed at the mRNA (Figure 3A) and protein (Figure 3B) levels in human hepatocellular carcinoma HepG2 and in human prostate cancer LNCaP (androgen-dependent) cells, whereas it was not significantly expressed in other types of cancer cell lines, including human prostate cancer PC-3 and DU-145. Thus, we propose that GK1 may play a role in the regulation of cancer cell death, although GK1 is well known as an intermediate enzyme between lipid and carbohydrate metabolism.

### 3.2. Expression, Enzymatic Activity, and Subcellular Localization of Exogenous Human Glycerol Kinase 1 Isoform b (GK1b) in Prostate Cancer PC-3 Cells

Our microarray data showed that human GK1 genes were upregulated by several anti-cancer agents. To validate the hypothesis that GK1 might have anti-tumorigenic effects, we designed human GK1 transcripts for exogenous encoding in PC-3 cancer cells using a pcDNA3.1/Hygro(+)-based gene expression method. The sequence of genes in the modified plasmid (Figure 1B) was identified to human GK1b. Identified DNA sequence can be found in Appendix A. To confirm the expression and functioning of recombinant GK1b, PC-3 cells were transfected with 0.3, 1, and 3 μg of recombinant GK1b for 48 h. GK1 was found to be overexpressed at the mRNA (Figure 4A) and protein (Figure 4B) levels post-transfection.

We also assessed the activity of intracellular GK1 in cells transfected with GK1b plasmid by the amount of glycerol-3-phosphate (Gro3P) produced. Intracellular Gro3P activity was significantly enhanced in GK1b-overexpressed PC-3 cells (PC3-GK1b) compared to PC3-Mock cells, which were transfected with an empty vector (Figure 4C). Furthermore, we confirmed that GK1 protein was primarily expressed in the cytoplasm (Figure 4D and Appendix A). This result is presumed to be due to the lack of a transmembrane domain in GK1 [26,27]. Overall, our results demonstrate that the transfected PC-3 cells adequately produced functioning GK1b at mRNA and protein levels.

### 3.3. Effect of Exogenously Overexpressed GK1b on Cell Viabilities, Morphological Alteration, and Cell Death of Prostate Cancer PC-3 Cells

We then sought to determine the role of GK1b in cell death. The cell viability of PC3-GK1b cells was significantly reduced by 60% at 72 h after transfection compared to vehicle (PC-3 parent) and PC3-Mock cells (Figure 5A). Also, GK1b induced an apoptotic-like morphology in transfected cells (Figure 5B). To clarify whether the PC3-GK1b cells were undergoing apoptosis, flow cytometry was performed on Annexin V and PI double-stained cells. GK1b clearly increased apoptosis dose-dependently compared to vehicle or PC3-Mock cells (Figure 5C). Moreover, PC3-GK1b upregulated the Bax/BCL-2 ratio, which determines the susceptibility of the cell to apoptosis, and cleaved caspase-3 and cleaved PARP, which are key indicators of apoptosis (Figure 5D). These findings suggest that GK1b plays a significant role as a suppressor gene during prostate carcinogenesis.

## 4. Discussion

Glycerol kinase (GK), also known as ATP-3-phosphotransferase, plays a pivotal role in cellular metabolism by catalyzing the conversion of glycerol to glycerol-3-phosphate (Gro3P) in a process requiring Mg^2+^ and ATP [28]. This process is essential as it provides the precursor for dihydroxyacetone phosphate, which is essential for glycolysis, gluconeogenesis, and glycerolipid metabolism. Thus, GK functions as an intermediary enzyme bridging carbohydrate and lipid metabolism. Conversely, Gro3P phosphatase (G3PP) hydrolyzes Gro3P to glycerol [29]. Pathologically, GK deficiency (GKD) has been associated with organic acidemia, glyceroluria, and hyperglycerolemia, which are often caused by mutations or deletions on the Xp21 of the X chromosome [30,31,32,33]. Furthermore, GK is implicated in insulin metabolism, which in its absence or mutation correlates with increased insulin resistance [11]. Interestingly, it has also been identified that GK has alternative functions and activities [34]; for example, GK interacts with the ATP-stimulated translocation protein, a molecule exhibiting homology with GK, which leads to the activation of the glucocorticoid receptor complex [35]. This complex interacts with histones, facilitating communication with porin or the voltage-dependent anion channels located on the outer mitochondrial membrane. This interaction triggers the translocation of Bax to the mitochondrial membrane, which is a crucial step in the regulation of apoptosis [36,37,38]. Moreover, GK overexpressed in rat hepatoma induces an excessive consumption of carbon sources [12]. In the context of disease, G3PP is highly expressed in prostate cancer patients with aggressive phenotypes [13], which is due to the survival response of cancer cells in a hostile environment characterized by glucolipotoxicity and metabolic stress induced by excessive nutrient availability [39,40]. Consequently, it can be speculated that GK overexpression leads to excessive carbon consumption and nutrient overload, which may potentially induce apoptosis in prostate cancer cells.

We investigated endogenous GK1 expression in various tissues and cell lines using ‘The Human Protein Atlas dataset’ (an open data resources) [41,42]. According to this database, GK1 is broadly expressed in normal tissues and expressed at robust levels in intestinal, renal, and hepatic tissues and cell lines. Although strong expression has been observed in certain cancer, such as hepatocellular carcinoma and thyroid cancer, the specificity of GK expression is notably low in most cancers. Additionally, moderate levels of GK positivity were reported in endometrial and prostate cancers. Remarkably, our analysis revealed that most androgen-dependent cell lines (including 22Rv1, LNCaP clone FGC, MDA-PCa-2b, and VCaP) expressed GK at higher levels than the androgen-independent cell lines DU-145 and PC-3, although the NCI-H660 cell-line proved to be an exception. Consistent with the open-source database, our experimental results demonstrated that the majority of cancer cells have lower GK expression at both mRNA and protein levels than hepatocellular carcinoma HepG2 cells (Figure 3). Additionally, GK1 was highly expressed at both mRNA and protein levels in LNCaP cells, which is consistent with the open-source database (Figure 3). It is unclear why GK is differently expressed in a subset of prostate cancer cells, though it is conceivable that GK may implicated in the regulation of sex hormones in prostate cancer. Lounis et al. have provided the evidence indicating that G3PP, which acts in the opposite manner to GK, is highly expressed in androgen-dependent cells, such as 22Rv1 and LNCaP, while it is expressed at low levels in androgen-independent cells such as PC-3, DU-145, and C4-2B [13]. Although the exact cause remains elusive, this suggests that GK may be closely related to sex hormone regulation.

Based on the microarray results, we hypothesize that GK1 could play a role as a novel regulator of prostate cancer cell death. Four randomly selected anti-cancer agents significantly increased GK1 expression at both mRNA and protein levels, which concurred with the DNA microarray results (Figure 2 and Table 2). This suggests that GK and cell death are linked by factors yet to be identified. To explore this hypothesis, we transfected a GK expression plasmid into human prostate cancer PC-3 cells, which express GK at low levels. The cell viabilities of GK1b-overexpressed PC-3 cells (PC3-GK1b) were dramatically lower than either vehicle or Mock-transfected PC-3 cells (PC3-Mock) at 72 h post-transfection (Figure 5A). Moreover, the morphology changes observed in PC3-GK1b resembled those of cells undergoing apoptosis (Figure 5B). Subsequent Annexin V and propidium iodide staining followed by flow cytometry confirmed apoptosis induction. The intensity of apoptosis was accelerated by increasing amount GK1b plasmid transfected into PC-3 cells (Figure 5C). In addition, transfection with GK1b activated key apoptotic proteins, including cleaved caspase-3, cleaved PARP, and an increased Bax/BCL-2 ratio (Figure 5D). These findings collectively suggest that GK1, particularly the GK1b isoform, has therapeutic potential for modulating apoptosis in prostate cancer.

However, despite these compelling results, the correlation between cancer and GK remains controversial. Zhou et al. have discovered that exosomal GK5 mRNA was significantly higher in the plasma of gefitinib-resistant adenocarcinoma patients than in the plasma of gefitinib-sensitive patients, and that the mRNA and protein levels of GK5 were significantly higher in gefitinib-resistant human non-small cell lung cancer PC9R and H1975 cells than in gefitinib-sensitive PC9 cells [43]. It was also revealed that mitochondrial damage, the activation of the caspase cascade, cell cycle arrest, and apoptosis were induced through the SREBP1/SCD1 signaling pathway in GK5-silenced PC9R cells [43]. In addition, GK has been proposed to be a prognostic predictor in esophageal carcinoma (ESCA) based on a Kaplan–Meier survival analysis of Cancer Genome Atlas (TCGA) data, which indicated that elevated GK expression is associated with poorer outcomes in ESCA patients [44].

Nevertheless, the function of GK in cancer remains incompletely understood, which is presumably due to limited research on GK, GK subtypes, and the relationship between GK and prostate cancer. Consequently, there is a critical imperative to conduct further investigations on the role of GK1 through in vivo and human studies, as well as testing with multiple prostate cancer cell lines besides PC-3 cells, to unravel its potential therapeutic implications in targeting cancer cells.

## Figures and Tables

**Figure 1 biomolecules-14-00997-f001:**
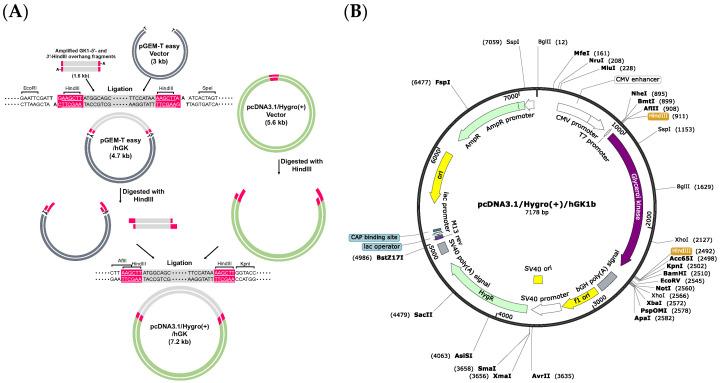
Construction scheme of an expression vector encoding human glycerol kinase 1 isoform b (GK1b). (**A**) Schematic drawing representing procedure for the establishment of human GK1b expression vector. For T-vector cloning, a pGEM^®^-T Easy system from Promega was executed. To express an exogenous human *GK1b* gene in mammalian cells, a pcDNA3.1/Hygro(+) vector was used; (**B**) the vector map of pcDNA3.1/Hygro(+)/GK1b.

**Figure 2 biomolecules-14-00997-f002:**
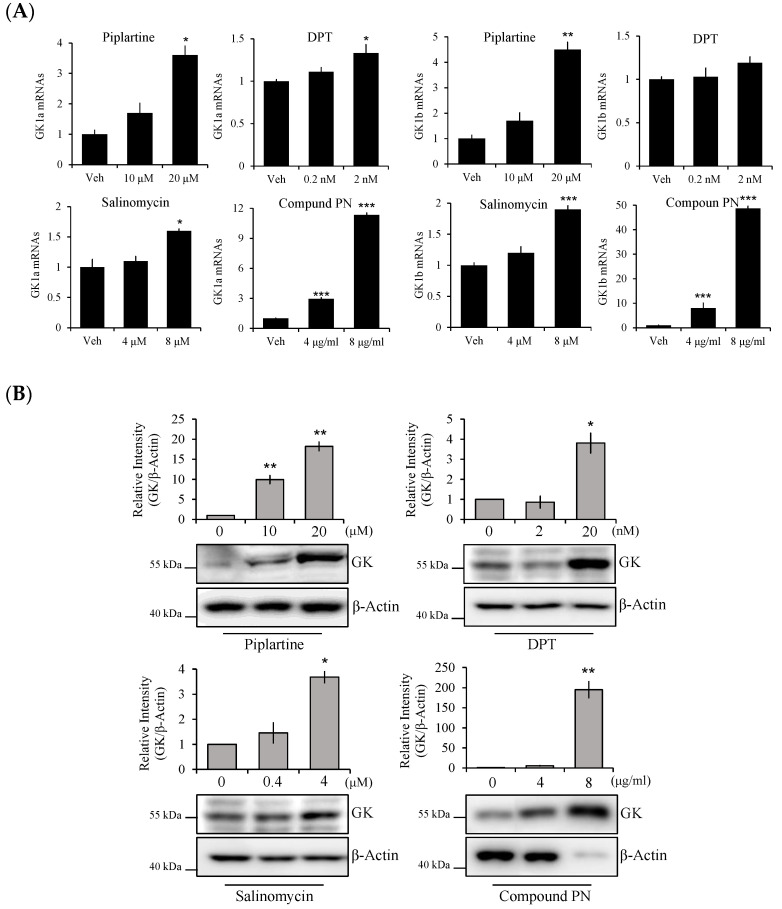
mRNA and protein expression of human GK1 by anti-cancer agents. (**A**) relative mRNA levels of glycerol kinase 1 isoform a (GK1a) and GK1b; (**B**) the protein levels of GK1. Human prostate cancer PC-3 cells were treated with piplartine, deoxypodophyllotoxin (DPT), salinomycin and compound K for 24 h, respectively. Process of RNA isolation and real time-qPCR analysis and Western blotting was described in the Section 2. And GK1a and GK1b mRNAs were normalized with GAPDH as a reference and quantified by the 2^−ΔΔCt^ method. Protein expression was quantified with ImageJ and normalized to β-Actin. All data are indicated as mean ± SEM (*n* = 3 in each group). * *p* < 0.05, ** *p* < 0.01 and *** *p* < 0.001 vs. the vehicle group. Vehicle, sample dissolved-solvent treated group.

**Figure 3 biomolecules-14-00997-f003:**
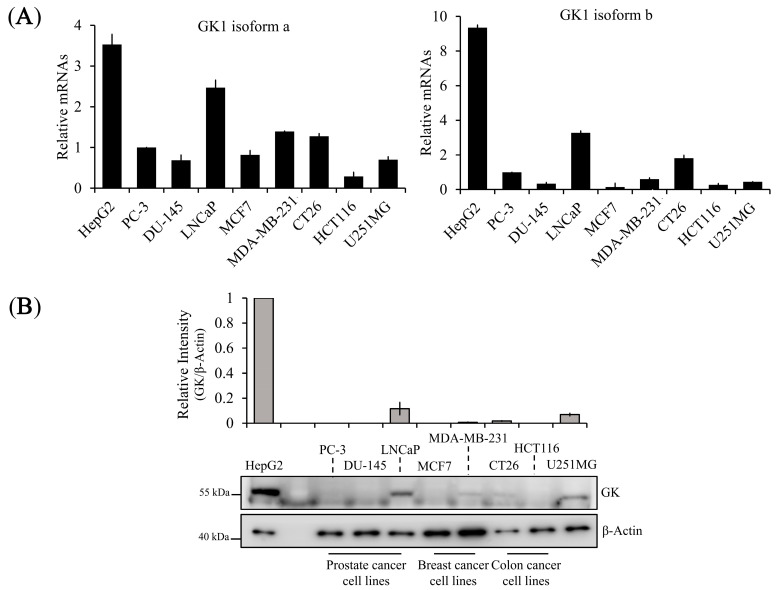
mRNA and protein expression of endogenous GK1 in multiple cancer cell lines. (**A**) mRNA expression of GK1a and GK1b isoforms in multiple cancer cell lines; (**B**) protein expression of GK1 in multiple cancer cell lines. Protein expression was quantified with ImageJ and normalized to β-Actin. All data are indicated as mean ± SEM (*n* = 3 in each group).

**Figure 4 biomolecules-14-00997-f004:**
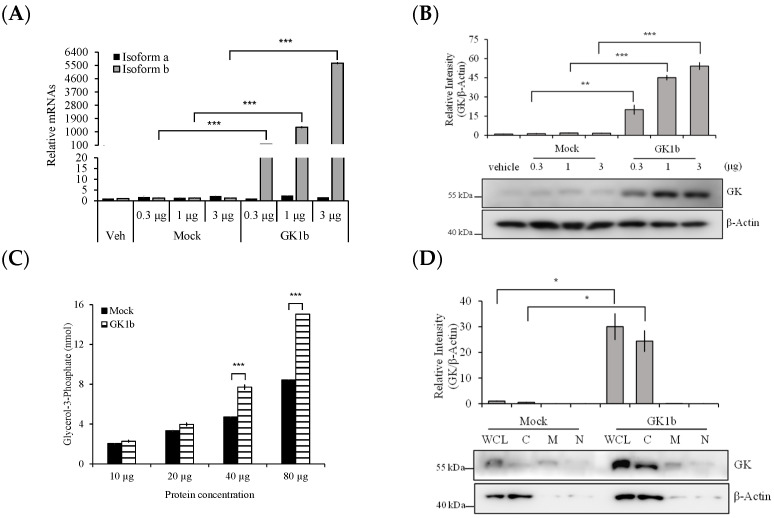
Expression, enzymatic activity and subcellular localization of exogenous human GK1b in prostate cancer PC-3 cells. (**A**) mRNA levels of GK1a and GK1b isoforms. GK1b mRNA was only increased in a dose-dependent manner; (**B**) protein levels of exogenous GK1. GK1 was highly enhanced in a dose-dependent manner; (**C**) production of glycerol-3-phosphate (Gro3P) which is an indicator of GK activity. Protocol for measuring of the Gro3P production was described in the Section 2; (**D**) subcellular fractionation of GK1 in human prostate cancer PC-3 cells. Fractionation process was described in the Section 2. Protein expression was quantified with ImageJ and normalized to β-Actin. All data are presented as mean ± SEM (*n* = 3 in each group). * *p* < 0.5, ** *p* < 0.01, and *** *p* < 0.001 vs. Mock. Vehicle, transfection reagent treated group; Mock, empty vector (pcDNA3.1/Hygro(+)) transfected group; GK1b, pcDNA3.1/Hygro(+)/GK1b transfected group; WCL, whole cell lysate; C, cytosol fraction; M, membrane-bound organelles fraction; N, nuclear fraction.

**Figure 5 biomolecules-14-00997-f005:**
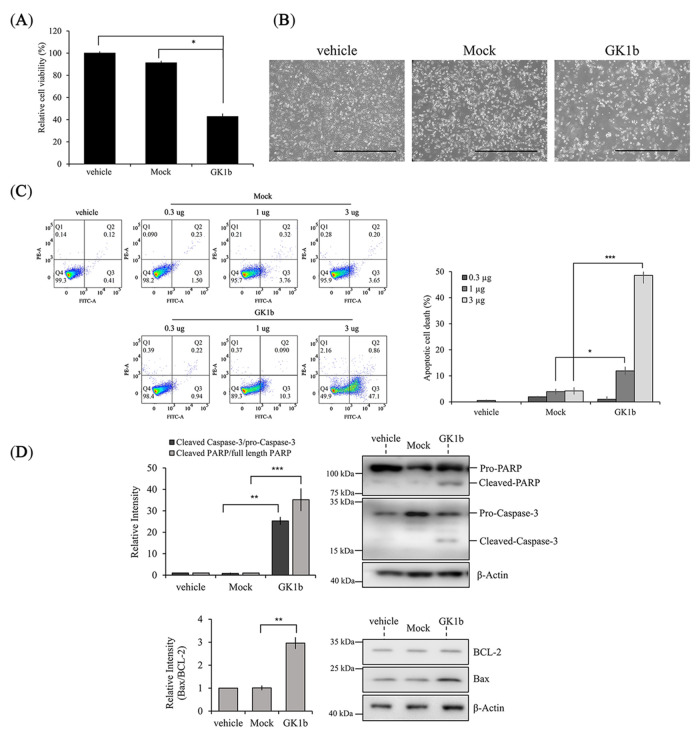
Expression of exogenous human GK1b and its effects on cell viability and morphological alteration in prostate cancer cells. (**A**) Cell viability of PC-3 cells after transfection for 72 h; (**B**) cell morphology of PC-3 cells after transfection. Representative images obtained by optical microscope after employing recombinant GK1b for 48 h. Scale bar is 1000 µm; (**C**) dot plot and quantitative graph for apoptosis; (**D**) apoptosis-inducible proteins. Cells were transfected with various concentrations of recombinant GK1b for 48 or 72 h. Protein expression was quantified with ImageJ and normalized to pro-casepase-3, full-length PARP, and β-Actin, respectively. All data are presented as mean ± SEM (*n* = 3 in each group). ^¶^ *p* < 0.05 vs. vehicle and * *p* < 0.05, ** *p* < 0.01, and *** *p* < 0.001 vs. Mock. vehicle, transfection reagent treated group; Mock, empty vector (pcDNA3.1/Hygro(+)) transfected group; GK1b, pcDNA3.1/Hygro(+)/GK1b transfected group.

**Table 1 biomolecules-14-00997-t001:** Classification of human glycerol kinase (GK) subtypes.

Name	Synonym	Accession Number	Location
*Homo sapiens* GK, transcript variant 1, mRNA	GK1 isoform a	NM_203391.4	X Chromosome
*Homo sapiens* GK, transcript variant 2, mRNA	GK1 isoform b	NM_000167.6	X Chromosome
*Homo sapiens* GK, transcript variant 3, mRNA	GK1 isoform c	NM_001128127.3	X Chromosome
*Homo sapiens* GK, transcript variant 4, mRNA	GK1 isoform d	NM_001205019.2	X Chromosome
*Homo sapiens* GK2, mRNA	GK2	NM_033214.3	Chromosome 4
*Homo sapiens* GK3, mRNA	GK3	NM_001395953.1	Chromosome 4
*Homo sapiens* GK5, transcript variant 1, mRNA	GK5	NM_001039547.3	Chromosome 3

These data were collected from the National Center for Biotechnology Information “https://www.ncbi.nlm.nih.gov/ (accessed on 13 June 2024)” [14] and UniProt “https://www.uniprot.org/ (accessed on 13 June 2024)” [15], which are open resources to freely access the high-quality of information regarding nucleotide accession numbers, protein sequences and their locations and functions. This table presents the GK subtype’s names, synonyms, accession numbers, and chromosomal locations.

**Table 2 biomolecules-14-00997-t002:** Microarray profiles of human GK1 isoforms by anti-cancer agents.

Gene	Glycerol Kinase Isoform b	Glycerol Kinase Isoform a	Glycerol Kinase Isoform a	Glycerol Kinase Isoform a
Systematic	A_32_P172848	A_24_P100387	A_24_P100382	A_23_P96556
RefSeq	NM_000167	NM_203391	NM_203391	NM_203391
Con vs. Comk	2.8196.94	3.2657862	3.9736977	2.5450099
Con vs. PD	3.264055	3.7880125	3.8311794	2.5796342
Con vs. P18	4.4688272	3.1760504	3.8785555	3.167038
Con vs. P20	1.7412026	2.563789	3.0455835	1.6804707
Con vs. P22	2.6428287	3.4274137	3.5580418	2.890148
Con vs. P26	6.13855	5.7897286	6.1781945	5.036399
Con vs. P36	4.9438443	6.9973483	8.964015	6.3650146

This table represented the relative fold-change expression of the *GK1* gene after treatment of seven different anti-cancer agents. The whole human genome microarray (4 × 44 K) kit used in this study contains four probes that detect GK1 isoform a (GK1a) and GK1 isoform b (GK1b) transcripts. Three of these, A_23_P96556, A_24_P100382, and A_24_P100387, target different sequences in GK1a. Con, control; Comk, compound K; PD, platycodin D; P18, lasalocid; P20, deoxypodophyllotoxin (DPT); P22, piplartine; P26, salinomycin; P36, pipernonaline.

## Data Availability

The original contributions presented in the study are included in the article/Appendix A; further inquiries can be directed to the corresponding authors. Endogenous glycerol kinase (GK) expression in multiple tissues and cell lines data utilized in this study were obtained from The Human Protein Atlas and are cited in [41,42]. Table 2 and Appendix A were created using the database acquired from the National Center for Biotechnology Information and UniProt and cited in [14,15].

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
