# Peer review of "Exploring a Novel Role of Glycerol Kinase 1 in Prostate Cancer PC-3 Cells"

_biomolecules, 2024, doi:10.3390/biom14080997_

Round 1

Reviewer 1 Report

Comments and Suggestions for Authors

The passage outlines a significant discovery regarding glycerol kinase 1 (GK1) in the context of prostate cancer research. The discovery of GK1's involvement in promoting apoptotic cell death in prostate cancer cells, particularly through its isoform GK1b, represents a promising advancement in cancer research. It opens up new avenues for exploring GK1 as a therapeutic target and biomarker, potentially leading to more effective treatments for prostate cancer in the future. This is an interesting manuscript, but I have several following concerns:

1. I wonder to know the other functions of Glycerol Kinase 1 in Prostate Cancer 2 PC-3 Cells, such as migration, invasion and drug resistance. 

2. Abbreviations should be defined or written in full when they first appear. Such as "cRNA" in line 108, "MTT" in Line 242, "SEM" in Line 258..., please double check all the text to find similar errors and correct them.

3. All the Figures in the text should be in the same pages and should not span pages..

4. You should add a scale bar for the Figure 4B. 

5. In Line 154, "MDAMB231" should be "MDA-MB-231", please double check all the names of cell lines and correct them using standard writing method.

6. Please quantify and statistically analyze the WB results in Figure 2-5.

7. The nucleic acid sequences (including gene names, regulatory sequences, and primer names) should be in italics.

8. Tables should use a standard three-line table and should not span pages.

9. Please unify the format of references in the article, including the author's name, the case of words in the title of the article, the writing of the name of the journal, and the page number.

Comments on the Quality of English Language

Moderate editing of English language required.

Reviewer 2 Report

Comments and Suggestions for Authors

Authors in this studied the role of GK1 in promoting cell death in prostate cancer cells. Although the authors did lot of experiments and produced some convincing data, there are several issues in the manuscript which should be addresses before it can be accepted for publication. All my comments which should be addressed are given below:

1.        Line 104 in material and methods section, it should be processes instead of process. 

2.   Line 110-111- This sentence should be rephrased. “After that, instantly cool on ice and add 2x Hybridization Buffer to stop this reaction”. Write it in the past tense.

3.   “Agilent’s GeneSpring Software (GebeSpringGX 7.3.1) was performed to normalize and cluster of our all raw data to sort by fold-change of genes”. This whole sentence should be rephrased.

4.   “Synthesized cDNA was performed qPCR analysis with Real-Time qPCR 2x Master Mix (ELPIS) by using ABI QuantStudio3 machine (Applied Biosystems,Waltham, MA, USA)”. This sentence should be rephrased.

5.   Apart from the mistakes mentioned in above comments, there are several other mistakes in the manuscript related to English language. I would recommend the authors should consider having the manuscript edited by professional English editor.

6.   What are the three values in table 2. for Glycerol kinase isoform a? Are these triplicate values? Why there is only one value for Glycerol kinase isoform b?

7. The actin bot in Figure B is overprocessed. Authors should replace it.

8. Is there any specific reason that authors used different concentration of Salinomycin and DPT for q-PCR and WB?

9. How many times these WB experiments were done?

10.  What about the effect of drug treatment on other cell models? Is the drug effect on GK at protein level and on the mRNA levels of GK isoform a and GK isoform b specific to PC3 cells? 

11.  In figure 4B, the blot is probed with GK antibody. Does this antibody differentiate between isoforms? Authors should explain this. In this blot is it the expression of GK isoform b or it is GK?

12.  In FIGUR 4A and Figure 4B: After transfection of plasmid expressing GK isoform b at mRNA level there is huge increase when compared to 0.3ug and 1ug, but at protein level it’s not the case. Authors should explain it in the text.

13.  In addition to WB, I would recommend authors to perform IF and show that the expression of GK1B is predominantly in cytoplasm.

14.  Figure 5B: There is no scale bar. Authors should include it.

15.  All the experiments in Figure 5 were conducted in PC3 cells. Did the authors perform these experiments in other cell lines? If not, why? Is this phenotype specific to only PC3 cells? If yes, why? Authors should explain this.

16.  In figure 5E: there is no change in BCL-2. Authors should also explain this in the text. 

17.  How many times experiments in Figure 5D were done? Authors should normalize the values for each protein with loading control and include the graph with statistics.

18.  Authors should include the catalog number of all the antibodies used in the study.

Comments on the Quality of English Language

There are several mistakes. Manuscript should be edited by professional English editor.

Round 2

Reviewer 1 Report

Comments and Suggestions for Authors

The authors have addressed all my concerns. I recommend accepting it in current form. 

Author Response

We appreciate for all your comments and suggestions that have made the manuscript stronger.

Reviewer 2 Report

Comments and Suggestions for Authors

Thanks for addressing all my comments. There are few minor corrections which should be addressed. They are as follows.

1. In Line 48 where authors discuss epigenetic and other modifications- Apart from reference 8 also this reference should be included- “Phenotypic plasticity- Alternate transcriptional programs driving treatment resistant prostate cancer”. Critical reviews in oncogenesis. 2022.

2. In Line 360: I would suggest authors also include this recent reference for symptoms of Glycerol deficiency - “Korkut, S. et al. Complex glycerol kinase deficiency and adrenocortical insufficiency in two neonates.” J. Clin. Res. Pediatr. Endocrinol. 2016.

3. In figure 1A: The sequences presented by authors are not readable. They should increase the font size of these sequences so that they are visible. 

4. In figure 5D: The Y axis labeling is merging into the numbers. Space should be added there so that they don’t merge.

5. In figure 2A: The graphs are not properly aligned. They should be aligned properly.

6.In Figure 2B: The spacing between western blotting images is not homogeneous. In the top right WB images the blots are almost merging. Space should be added to separate them.

Author Response

  1. In Line 48 where authors discuss epigenetic and other modifications- Apart from reference 8 also this reference should be included- “Phenotypic plasticity- Alternate transcriptional programs driving treatment resistant prostate cancer”. Critical reviews in oncogenesis. 2022.
    • Author response: We appreciate for all your comments and suggestions that have made the manuscript stronger. We’ve cited the paper you suggested in the manuscript.
  1. In Line 360: I would suggest authors also include this recent reference for symptoms of Glycerol deficiency - “Korkut, S. et al. Complex glycerol kinase deficiency and adrenocortical insufficiency in two neonates.” J. Clin. Res. Pediatr. Endocrinol. 2016.
    • Author response: We appreciate for all your comments and suggestions that have made the manuscript stronger. We’ve cited the paper you suggested in the manuscript.
  1. In figure 1A: The sequences presented by authors are not readable. They should increase the font size of these sequences so that they are visible. 
    • Author response: Thank you for your comment. We’ve addressed the issue you mentioned.
  1. In figure 5D: The Y axis labeling is merging into the numbers. Space should be added there so that they don’t merge.
    • Author response: Thank you for your comment. We’ve addressed the issue you mentioned.
  1. In figure 2A: The graphs are not properly aligned. They should be aligned properly.
    • Author response: Thank you for your comment. We’ve addressed the issue you mentioned.
  1. In Figure 2B: The spacing between western blotting images is not homogeneous. In the top right WB images the blots are almost merging. Space should be added to separate them.
    • Author response: Thank you for your comment. We’ve addressed the issue you mentioned.